# Demographic history and biologically relevant genetic variation of Native Mexicans inferred from whole-genome sequencing

Sandra Romero-Hidalgo [1], Adrián Ochoa-Leyva[1,2], Alejandro Garcíarrubio[2], Victor Acuña-Alonzo[3], Erika Antúnez-Argüelles[1], Martha Balcazar-Quintero[4], Rodrigo Barquera-Lozano [3], Alessandra Carnevale[1], Fernanda Cornejo-Granados[2], Juan Carlos Fernández-López[1], Rodrigo García-Herrera [1], Humberto García-Ortíz[1], Ángeles Granados-Silvestre[5], Julio Granados[6], Fernando Guerrero-Romero[7], Enrique Hernández-Lemus [1], Paola León-Mimila[1,5], Gastón Macín-Pérez[3], Angélica Martínez-Hernández[1], Marta Menjivar[1,5,8], Enrique Morett[1], Lorena Orozco[1], Guadalupe Ortíz-López[8], Fernando Pérez-Villatoro[1,9], Javier Rivera-Morales[3], Fernando Riveros-McKay[1,9], Marisela Villalobos-Comparán[1], Hugo Villamil-Ramírez[1,5], Teresa Villarreal-Molina[1], Samuel Canizales-Quinteros[1,5] & Xavier Soberón[1,2]

Understanding the genetic structure of Native American populations is important to clarify their diversity, demographic history, and to identify genetic factors relevant for biomedical traits. Here, we show a demographic history reconstruction from 12 Native American whole genomes belonging to six distinct ethnic groups representing the three main described genetic clusters of Mexico (Northern, Southern, and Maya). Effective population size estimates of all Native American groups remained below 2,000 individuals for up to 10,000 years ago. The proportion of missense variants predicted as damaging is higher for undescribed (~ 30%) than for previously reported variants (~ 15%). Several variants previously associated with biological traits are highly frequent in the Native American genomes. These findings suggest that the demographic and adaptive processes that occurred in these groups shaped their genetic architecture and could have implications in biological processes of the Native Americans and Mestizos of today.

[1] Instituto Nacional de Medicina Genómica (INMEGEN), Mexico City 14610, Mexico. [2] Instituto de Biotecnología, Universidad Nacional Autónoma de México (UNAM), Cuernavaca, Morelos 62210, Mexico. [3] Escuela Nacional de Antropología e Historia, Mexico City 14030, Mexico. [4] Dirección de Formación y Acción Social, Universidad Iberoamericana, Mexico City 01298, Mexico. [5] Facultad de Química, UNAM, Mexico City 04510, Mexico. [6] Instituto Nacional de Ciencias Médicas y Nutrición Salvador Zubirán, Mexico City 14080, Mexico. [7] Unidad de Investigación Biomédica, Instituto Mexicano del Seguro Social, Durango 34067, Mexico. [8] Hospital Juárez de México, Mexico City 07760, Mexico. [9] Winter Genomics, Mexico City 07300, Mexico. Sandra Romero-Hidalgo, Adrián Ochoa-Leyva and Alejandro Garcíarrubio contributed equally to this work. Correspondence and requests for materials should be addressed to S.C.-Q. (email: cani@unam.mx) or to X.S. (email: xsoberon@inmegen.gob.mx)

The peopling of the Americas was the most recent continental occupation to occur in human history. Although still a matter of controversy, according to the most widely accepted model based on archaeological and genetic evidence, American Natives originated in Eastern Asia no earlier than 23 thousand years ago (kya), reached America through the Bering Strait and expanded across the Americas along the North–South direction[1, 2]. The inhabitants of the Americas of today are the result of several ongoing migration, admixture, and adaptive processes that have varied throughout the continent over several centuries. In the fifteenth century, prior to the Conquest of Mexico, Mexican territory was inhabited by many different Native groups located mainly in Mesoamerica, but also in Northern Mexico, inhabited by nomadic or semi-nomadic peoples[3]. The Mexican population of today is the result of complex and ongoing admixture processes that began with the Conquest of Mexican territory by the Spaniards in 1521[4]. The admixture process occurred mainly between Native Americans and Europeans, although there was a smaller contribution of the African population introduced by slave traders during the colonial period[5, 6]. Currently, there are 68 acknowledged Native American languages in Mexico. Native populations not only contributed very importantly to the admixture process of the Mexican population, but currently up to 21% of Mexicans identify themselves as members of one of the acknowledged Indigenous groups of Mexico. Moreover, according to a recent survey, approximately 7% of the Mexican population, that is over 7 million Mexicans, speak a Native language[7].

The genetic structure of Native Americans (NAs) of Mexico has been previously analyzed using microarray technology, shedding light on peopling and migration processes, and more recently on the high diversity of the NA populations[1, 8]. Much of what we know comes from analyzing patterns of common, and, therefore, ancient genetic polymorphisms via genotyping across diverse human populations[9, 10]. Recent studies have used sequencing approaches to reveal a more complete and genome-wide picture of variation, including low frequency variants of a more recent origin[11, 12]. However, only a few whole genome sequences from NA individuals have been reported, and the genetic variation of NA remains largely unexplored[2, 13, 14].

Understanding human genomic variation is a central focus of medical and population genomics[8, 15]. Thus, to better understand the genetic variability of NA and the genetic factors that may contribute to biological traits in this population, high coverage whole-genome sequences of 12 NA individuals from six distinct ethnic groups and a trio of Mexican Mestizos were analyzed, together with microarray data from 312 NA individuals. These data shed light on recent demographic history, better define the diversity of the populations of Mexico, and discover genetic variation shared among these populations, which may have an impact on health and other phenotypic traits.

## Results
**Sample selection.** Whole genomes of a total of 15 Mexican individuals were sequenced, including 12 NAs from six distinct ethnic groups (Tarahumara and Tepehuano from the North; Nahua, Totonaca, Zapoteca from the South; and Maya from the South-East) and a trio of Mexican Mestizos (mother, father, and offspring) (Fig. 1). NA participants were selected considering ancestry, linguistic group, geographic location, and representation of three of the main genetic clusters previously described in the NA Map of Mexico: Northern, Southern, and Mayan, as described by Moreno-Estrada et al.[8] Estimated NA ancestry using genome-wide data was over 98% in all indigenous participants, except for both Tepehuanos who were selected for having the

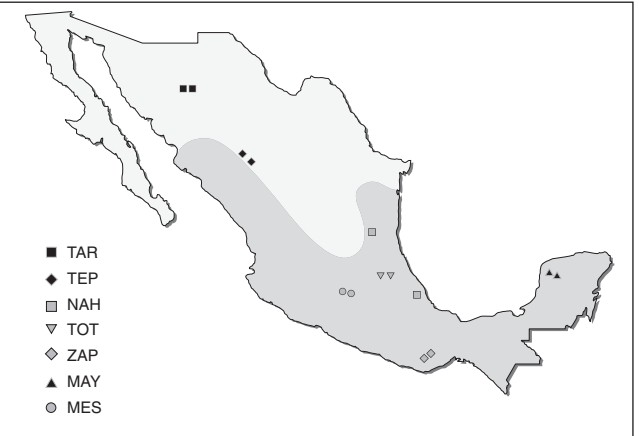

**Fig. 1** Map of Mexico. The figure shows the geographical origin of each NA and unrelated Mestizo individuals included in the present study. The region of Mesoamerica is shaded in gray. TAR (Tarahumara), TEP (Tepehuano), NAH (Nahua), TOT (Totonaca), ZAP (Zapoteca), MAY (Maya), and MES (Mestizo)

highest NA ancestry (91%) among those available for whole-genome sequencing (WGS).

**Analysis of population admixture.** Reference NA and continental populations were used to generate multidimensional scaling (MDS) plots. Figure 2a shows that components 1 and 2 distinguish Africans and Europeans from NAs of Mexico, and the 12 NA individuals clustered within the latter group. Moreover, component 3 separates NAs, showing the 12 NA individuals clustered within 3 of the main NA components described by Moreno-Estrada et al.[8] (Fig. 2b). Three-hundred and twelve additional samples from Nahua, Totonaca, Zapoteca, and Maya individuals were genotyped and included in the admixture analysis, revealing the presence of two additional and distinct Southern components: Nahua and Totonaca (Fig. 2c). Thus, the previously described Southern component includes at least three distinctive ancestral components: Nahua, Totonaca, and Zapoteca. Figure 2d shows ancestry proportions of the 12 NAs. The Tarahumara and Tepehuano individuals showed the highest average proportions of the Northern native component, while Mayas and Zapotecas showed higher proportions of their respective reference components. Interestingly, admixture analyses revealed that the four Nahua and Totonaca individuals show a distinct Native substructure. The high demographic and genetic diversity of Nahua-speaking populations is most likely a consequence of the expansion of the Aztec Empire mainly during the Postclassic period (1427–1520 A.D.)[16, 17]. Because the inclusion of a greater number of Nahuas and Totonacas in the analysis revealed the presence of previously unidentified components, and there are 1.7 million Nahuatl-speaking individuals (the largest indigenous population in Mexico)[7], it is necessary to obtain genomic data from different Nahua populations to identify whether Nahuatl-speaking individuals share the Nahua component here identified.

**Sequence variation of Native Mexican and Mestizo individuals.** The mean coverage for all genomes was 40× (Supplementary Table 1). On average, 96.8% of the genome was called for both alleles, and 99.4% of the bases in the reference genome (build 37.2) were covered by at least one read. The mean number of single-nucleotide variants (SNVs) was 3.2 million for the 12 NAs and 3.4 million for the three Mexican Mestizos, while the mean percentage of novel SNVs was 1.76% (Supplementary

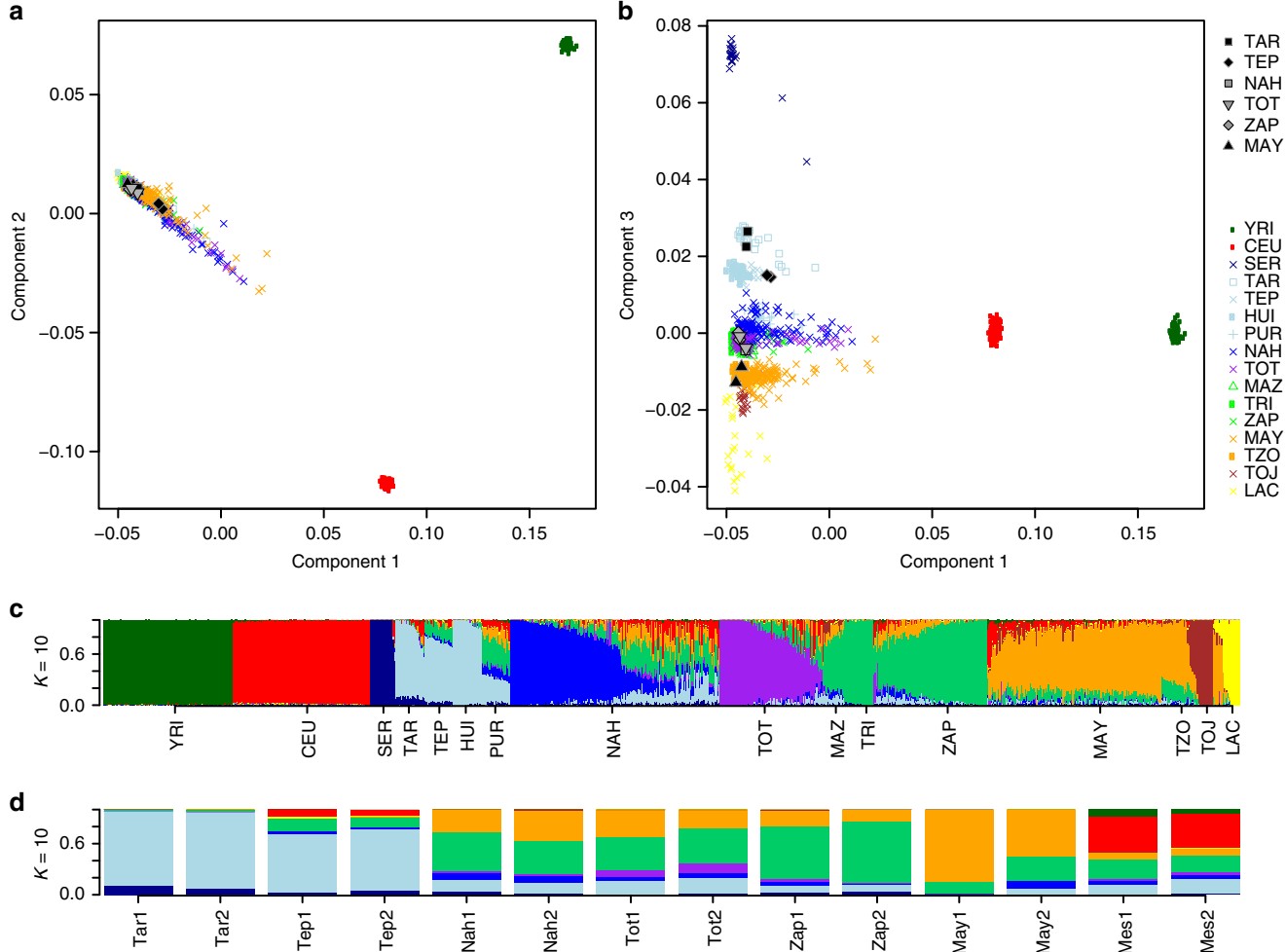

**Fig. 2** Multidimensional scaling plots and admixture analysis. **a** MDS plot for components 1 and 2 of 12 Native Americans of Mexico combined with continental (CEU and YRI from HapMap) and NA reference populations. **b** MDS plot for components 1 and 3, separating the NA populations of Mexico. The 12 sequenced NA samples are shown in gray, and population labels are as described in Supplementary Table 8. **c** Global ancestry proportions of NA and continental reference populations assuming $K = 10$. **d** Global ancestry proportions of the 12 NA individuals assuming $K = 10$. The NA individuals are displayed North-to-South, and the Mestizo individuals are displayed at the far right. Tar1 and Tar2 (Tarahumara), Tep1 and Tep2 (Tepehuano), Nah1 and Nah2 (Nahua), Tot1 and Tot2 (Totonaca), Zap1 and Zap2 (Zapoteca), May1 and May2 (Maya), Mes1 and Mes2 (Mestizo)

Table 2). Overall, 0.62% SNVs were located in coding regions, 0.28% were missense, and 0.002% were nonsense. Interestingly, the proportion of novel missense variants predicted to be damaging was higher for novel (30%) than for all missense variants (15%) (Supplementary Table 3). This is consistent with previous observations of an increased proportion of deleterious vs. neutral variation in Mexican-American individuals, compatible with population bottlenecks possibly experienced by NAs[18, 19]. The distribution of insertions/deletions (indels), copy number variations (CNVs) and mobile element insertions (MEIs) are described in Supplementary Tables 4 and 5.

The mean percentage of SNVs in heterozygosis was similar among NAs, but on average was higher in the Mestizo genomes (59.9%) as compared to the NA genomes (51.8%) (Supplementary Table 6). Considering the three previously described clusters, mean SNV heterozygosis was similar in Northern groups (Tarahumaras and Tepehuanos, 51.4%) Southern groups (Nahuas, Zapotecas, and Totonacas, 51.7%) and Mayas (52.5%). Relatedness between pairs of individuals as assessed by kinship coefficient was highest between both Totonacas (0.0639), suggesting they are third degree relatives. Estimated kinship coefficient between both Tepehuanos was 0.0005, and was zero for Mayas, Nahuas, Zapotecos, and Tarahumaras.

We then compared the local density of SNVs across the genome between the 12 NA and 1000 Genomes (1KG) populations excluding the Americas (AMR) populations (Supplementary Fig. 1). Overall, SNV density was similar in both populations. Notably, chromosome 6 showed three high-density peaks shared by both populations, and one peak found only in the 12 NA genomes. The latter peak includes the *PRIM2* gene, which encodes the 58 KDa subunit of DNA primase that plays a key role in DNA replication. This finding is consistent with that observed by Zhang et al.[20], who sequenced 35 Korean genomes finding that the *PRIM2* gene had an extremely high number of SNVs not found in the 1KG project.

**Inference of population history from sequence data.** As shown in Table 1, all five male indigenous participants shared the same Y-chromosome haplogroup (Q1a2a) distinctive of NA populations[21], and the male Mestizo had the R1b1 Y chromosome haplogroup, common in the Iberian Peninsula[22]. As expected, all individuals, NA and Mestizos, showed common NA mitochondrial DNA (mtDNA) haplogroups[23, 24]. Thus, these uniparental marker analyses confirm the demographic history of the Mexican Mestizo and Native populations, where the

**Table 1 Characteristics of Native American and Mestizo individuals**

| ID | Population | Location of origin | Linguistic group | Y chromosome | mtDNA haplogroup |
|----|-----------|-------------------|------------------|--------------|------------------|
| Tar1 | Tarahumara | Northern Mexico | Uto-Aztecan | Q1a2a1b | C |
| Tar2 | | | | — | C1c1a |
| Tep1 | Tepehuano | Northern Mexico | Uto-Aztecan | Q1a2a1a1 | C1b10 |
| Tep2 | | | | Q1a2a1a1 | A2c |
| Nah1 | Nahua | Central Mexico | Uto-Aztecan | Q1a2a1a1 | B2 |
| Nah2 | | | | — | A2B |
| Tot1 | Totonaca | Central Mexico | Totonac | Q1a2a1a1 | C1c2 |
| Tot2 | | | | — | A2u |
| Zap1 | Zapoteca | Southwestern Mexico | Oto-Manguean | — | A2m |
| Zap2 | | | | — | A2 |
| May1 | Maya | Southeast Mexico | Mayan | — | A2 |
| May2 | | | | — | C |
| Mes1 | Mestizo | Central Mexico | Spanish | R1b1a2a1a2b1a1 | A2g |
| Mes2 | | | | — | C |

contribution of the European gene pool to the admixture process was mainly of male origin[25].

A maximum likelihood tree was generated to ascertain the population history of present day NA in relation to worldwide populations by Treemix (Supplementary Fig. 2) using sequencing data from the 12 NA genomes (Tarahumara, Tepehuano, Nahua, Totonaca, Zapoteca, and Maya), 11 genomes from worldwide populations[26], and 4 ancient individuals (Neanderthal, Denisovan, Anzick-1, and the Mal'ta child)[27, 28]. The inferred tree recapitulates the North to South differentiation gradients for the NA groups observed from the MDS analysis. Southern NA populations of Mexico are grouped in a major clade (Totonacas, Zapotecas, Nahuas, and Mayas), while Northern NA populations of Mexico (Tarahumaras and Tepehuanos) branch from the same initial split. In addition, this analysis confirms the gene flow from the Mal'ta lineage, (known to have genetic affinities of both European and NA populations), to the ancestors of all present day NAs[2, 29].

Demographic history reconstruction using a pairwise sequentially Markovian coalescent (PSMC) model showed evidence of a strong bottleneck in the European, Asian and NA populations, reaching the lowest effective population size ($N_e$) between 50–60 thousand years ago (kya) (Fig. 3a). Although European and Asian $N_e$ recovered afterwards, an extended period of low population size is observed in NAs with $N_e$ around 2000 individuals for up to 20 kya, in consistency with previous reports[30, 31]. This extended bottleneck is also consistent with estimates of the time that the NA ancestors crossed the Bering Strait and moved into America[2, 32].

Because PSMC can only infer population size estimates beyond 20 kya[31, 33], multiple sequentially Markovian coalescent (MSMC) analyses were performed combining four and eight phased haplotypes. Figure 3b depicts MSMC analysis combining two genomes per NA group, showing that the $N_e$ remained close to 2000 individuals in all groups for another 10 thousand years (10–20 kya). We then performed MSMC analysis combining four individuals from Northern NA groups (Tarahumaras and Tepehuanos), and four individuals from Southern NA groups (Nahuas and Zapotecas). Figure 3c shows the analysis of these eight haplotypes, allowing the estimation of $N_e$ up to 2 kya. $N_e$ of Southern NA groups increased constantly between 9 and 2 kya, in contrast with Northern groups who continued to decline from 9 to 4 kya. This is consistent with the previously reported higher effective population sizes for Southern and Mayas compared to Northern NAs of Mexico[34]. In addition, this population growth concurs with the time estimated for domestication of maize in Southwestern Mexico, approximately 6000–10,000 kya[35–37].

**Identification alleles of biomedical relevance**. We analyzed all NA genomes for the presence of potentially pathogenic alleles as defined by the publication of the American College of Medical Genetics and Genomics (ACMG)[38]. Seeking for incidental findings within the 56 genes recommended by the ACMG, we found a total of 90 non-synonymous variants in the 12 NA genomes. Sixty-eight of these variants were polymorphisms (MAF > 1% in at least one of the 1KG populations). The 22 remainder variants were absent from the 1KG project (phase 3), three of which are annotated in the ClinVar database. One individual carried a mutation in the *KCNH2* gene (rs199472885, *R312C*) previously found in a patient with congenital long QT syndrome;[39] another carried a *BRCA2* variant annotated as of uncertain clinical significance in ClinVar (rs80358606), and a third mutation (*T187S*) was identified in the *SCN5A* gene, in the same position as a loss of function mutation (*T187I*) found in a patient with Brugada syndrome[40]. Because the phenotypes of these individuals and the penetrance of the variants annotated as pathogenic are unknown, it is not possible to determine whether the 19 remainder variants predicted as deleterious or probably damaging are in fact pathogenic.

In the last decade, up to 2400 genome-wide association studies (GWAS) have been published reporting over 20,000 *loci* associated to many different phenotypes. The vast majority of the studies have been performed in populations of European descent and few *loci* have been replicated in the Mexican population using designs that simultaneously analyze a broad number of single-nucleotide polymorphisms (SNPs)[41–43]. Based on the NHGRI GWAS catalog, which contains all the SNP-trait associations with a significance threshold of $10^{-5}$, a total of 10,347 variants were present in at least one of the 12 genomes and 1,888 variants were shared among the 12 individuals. The latter shared variants were grouped based on their experimental factor ontology (EFO) and their frequencies were compared with other continental populations, particularly Europeans (CEU) and Asians (CHB and JPT) from 1KG project. Notably, variants whose frequency in the 12 genomes most differed from European or Asian populations have been previously associated with human morphology and clinical traits (Supplementary Fig. 3 and Supplementary Table 7). Some of these variants have been reported to be involved in natural selection. For instance, the frequency of a skin pigmentation variant (rs1834640 near the *SLC24A5* gene) most differed from Europeans, but was similar to that in Asians. Earlier studies have highlighted *SLC24A5* as one of the top candidate genes demonstrating evidence for positive selection in Europeans, suggesting that the fixation of the "A" allele is related with adaptive processes[44, 45]. Moreover, the allele frequency of the rs765132 "T" allele, previously associated with response to anti-tumor necrosis factor alpha therapy in

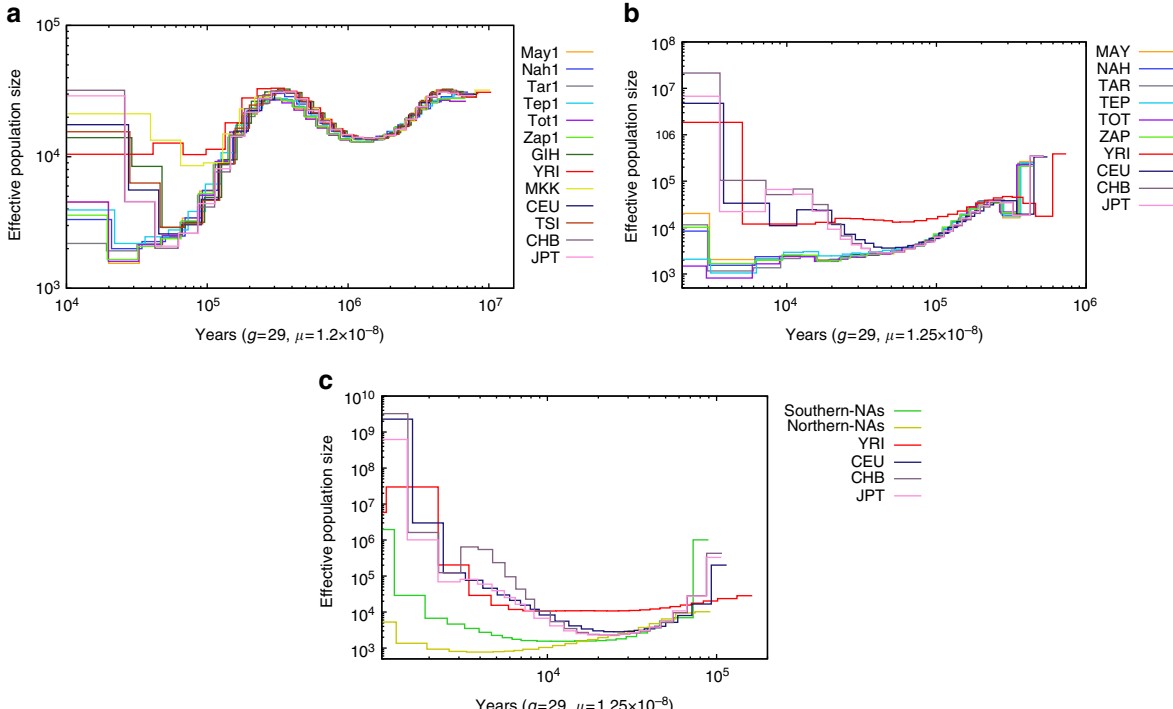

**Fig. 3** Inference of population sizes from whole-genome NA sequences. **a** Effective population size estimates from autosomes of six Native American (NA) individuals and an African, European, and Asian genome from the 1KG project, inferred using pairwise sequentially Markovian coalescent (PSMC) models. **b** Effective population size estimates from four haplotypes (two phased individuals from each of the six NA and 1KG populations), inferred using multiple sequentially Markovian coalescent (MSMC) models. **c** Effective population size estimates from eight haplotypes [four phased individuals from each of NA Northern (Tepehuanos and Tarahumaras), NA Southern (Nahuas and Zapotecas) and 1KG populations], inferred using MSMC models

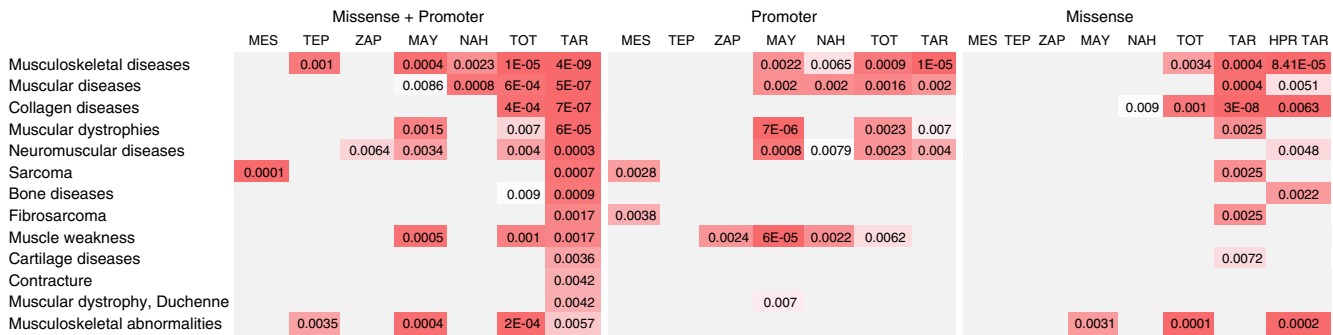

**Fig. 4** Enrichment analysis for disease-associated genes. Annotation matrix for disease-associated genes enriched in different populations shown as a heatmap, red indicates disease terms significantly enriched. Disease terms enrichment was analyzed by WebGestalt using the hypergeometric test for enrichment evaluation analysis. *P*-values were adjusted using Benjamini & Hochberg method. MES (Mestizo), TEP (Tepehuano), ZAP (Zapoteca), MAY (Maya), NAH (Nahua), TOT (Totonaca), TAR (Tarahumara), HPR TAR (high physical resistance)

inflammatory bowel disease[46], was 87.5% in the 12 NA whole genomes, 59% in the 312 additional NA samples with microarray genotyping data, but only 5% in Europeans (5%) and not found in Asians. Thirteen additional SNVs with high population allele frequency differences were also available in the 312 NA samples. Overall, allele frequencies of these SNVs were similar in the 12 NA whole genomes and the 312 NA samples, although differences ranged from 0.01 to 0.19 (Supplementary Table 7). Because of the potential clinical implications of these variants, it is important to design studies aiming to identify their biological implications in NAs and Mestizos of today.

**Pathway and gene ontology enrichment analysis.** Another approach to identify biologically relevant genes is to look not at

the variations of individual genes, but rather at gene sets or pathways[47]. We clustered the genes with novel non-synonymous and promoter region SNVs (Supplementary Table 2) in order to search for pathway and gene ontology (GO) enrichment. Supplementary Fig. 4 shows gene sets and/or pathways enriched with novel missense and promoter region mutations in the 12 NA individuals here analyzed. This analysis identified a wide range of enriched pathways in all groups (Supplementary Fig. 4a–g). Interestingly, one of the most significant enrichment values was for genes related to collagen, muscular and musculoskeletal diseases for both promoter and missense variants in Tarahumaras (Fig. 4). The combined analysis of these SNVs also showed highly significant enrichment for these pathways. This enrichment was also observed in other NA groups, although with less significant values. Moreover, to explore whether the genes

with novel missense mutations share specific functional features, we performed GO enrichment analysis by population. Interestingly, the only significantly enriched GO terms in Tarahumaras were related to molecular and cellular processes involved in musculoskeletal function (Supplementary Fig. 5), previously suggested to be involved in athletic performance[48, 49].

In order to seek further evidence of the enrichment in muscle and collagen-related pathways found in Tarahumaras, we sequenced the exome of three more individuals from this group, known to have participated in high resistance physical activity events. These individuals showed significant gene enrichment in muscle-related pathways, in consistency with the two initially sequenced Tarahumaras (Fig. 4). While these findings could be related with the well-known high physical resistance of this Native Mexican group[50, 51], they should be interpreted with caution. Further research including a larger sample size and phenotypic data should be performed to establish any possible relationship between gene enrichment and physical resistance in the Tarahumara population.

## Discussion

In conclusion, the study of NA genomes provides an enriching perspective on the demographic history of the region and is relevant in the genetic characterization of phenotypes in these populations. This study shows that the diversity of NA populations, particularly the Southern populations, is greater than what has been previously described[8]. Moreover, the demographic and adaptive processes suffered by these groups shaped their genetic architecture, having implications in biological and health-related processes of NAs and Mestizos of today. This is the first effort to characterize whole genomes of Native individuals from different regions of Mexico leading to the identification of variants associated with biological and clinical traits.

## Methods

**Ethics statement.** Institutional review board of the National Institute of Genomic Medicine (INMEGEN) approved this project. Written informed consent was obtained from all participants. For participants not fluent in Spanish, a translator was used. Blood samples were drawn with the permission of local authorities. The ethical duty to return individual research results or incidental findings is currently a matter of debate, depending on characteristics of the finding such as analytical and clinical validity, clinical actionability and significance as well as magnitude of potential damage. Because the samples included were donated anonymously, it was not possible to inform the participants about the incidental findings found here.

**Analysis of population admixture.** Genotype data from 112 CEU and 106 YRI from 1KG project; 401 NA samples from Moreno-Estrada et al.[8] (described in Supplementary Table 8) and 312 additional NA samples (103 Nahuas, 62 Totonacas, 49 Zapotecas, and 98 Mayas) genotyped with SNP 6.0 microarray (Affymetrix), were used for population admixture analysis. A total of 322,098 autosome-wide SNPs shared among these populations were used to generate MDS plots based on genome-wide identity by state pairwise distances as implemented in PLINK[52]. Supplementary Figs. 6 and 7 show the estimated individual ancestry proportions for $K = 3$ to $K = 10$ in 312 NA samples, continental reference populations, and the 12 NA and 3 Mexican Mestizo whole genomes using ADMIXTURE[53]. The fit of different values of $K$ was assessed using cross-validation (CV) procedures, where $K = 10$ showed the lowest CV error (Supplementary Fig. 8).

**Samples and sequencing procedure.** Sampling locations are described in Fig. 1. A total of 15 genomes (12 NAs from Tarahumara, Tepehuano, Nahua, Totonaca, Zapoteca, and Maya groups, and a trio of Mexican Mestizos) were submitted to WGS by Complete Genomics (Mountain View, California, USA). Individuals whose parents and grandparents recognized themselves as indigenous, had been born and lived in their home communities and spoke their native language were considered as NA. Individuals were selected according to their estimated NA ancestry, based on genome-wide data using the block relaxation algorithm implemented in ADMIXTURE[53], assuming $k = 3$ and including genotype data from CEU and YRI populations (1KG project) as well as additional NA samples included with SNP 6.0 microarray (Affymetrix) data.

All sequencing data were generated with Complete Genomics local pipeline[54]. Sequence reads were mapped to the reference human genome (GRCh37) and

variants were called using version 2.4 of the Complete Genomics software. All variant information (SNVs, indels, CNVs, MEIs) and structural variation] was obtained from the master variation files reported by Complete Genomics. Only variants classified as "PASS" were considered for analyses. Variants not reported in dbSNP v137 were considered novel. SNVs within the 7.5 kb region upstream of the 5′ transcription initiation site of genes were considered as promoter variants. The functional consequences of nonsynonymous SNVs were predicted using PolyPhen-2 (Polymorphism Phenotyping v2)[55]. The mean concordance rate between SNP 6.0 microarray and WGS data was 99.2% (Supplementary Table 9). Six selected variants found in COL4A2, COL5A2, and COL18A1 genes in Tarahumaras were confirmed by Sanger sequencing.

Relatedness between among the 12 NA individuals was inferred by the kinship coefficient using KING v2.0[56] from 4.8 million autosomal SNVs fully called in the 15 whole genomes (12 NA and the Mestizo trio), and biallelic for at least one genome. Kinship coefficients between parents and offspring of the trio were 0.251 and 0.253.

**Whole-exome sequencing.** Samples of three additional Tarahumaras known to have participated in high resistance physical activity events were prepared using Agilent SureSelect V4 All Exome kit (Agilent, Santa Clara, CA) and were sequenced on an Illumina NextSeq 500 system (Illumina, San Diego, CA) at the Sequencing Unit of INMEGEN. Reads were mapped to the reference human genome (GRCh37) with BWA[57], GATK was used for variant calling[58], and snpEff was used for variant annotation[59].

**Haplogroup assignment.** mtDNA sequences were compared to the revised Cambridge Reference Sequence[60] using Phylotree[61] and the nomenclature adopted by the International Society for Forensic Genetics[62]. Haplogrep software was used to assign haplogroups[63]. Y chromosome sequences were merged with a set of 19 worldwide Y chromosomes made publicly available by Complete Genomics, in order to determine their broad haplogroup affiliation.

**Genome-wide SNVs density.** The distribution of SNVs along genomes was analyzed dividing the genome into 20 Kb windows, and reporting the mean number of SNVs per window in the 12 NA genomes and in all non-AMR individuals from the 1KG project.

**Maximum likelihood analysis.** History of population splits and mixtures was inferred with TreeMix[64]. A total of 21 individuals were used to infer admixture graphs, 17 present-day individuals including the 12 NA genomes sequenced in this study, and 4 ancient individuals (Neanderthal, Denisovan, Anzick-1, and the Mal'ta child)[26–28]. Transitions were excluded from the data set to prevent biases introduced by the four ancient genomes. Only SNVs called in all genomes were included in the analysis. A maximum likelihood tree was inferred including 336,272 autosomal SNVs, using the default parameters and fitted allowing two migration edges.

**Inference of population history from sequence data.** Firstly, a pairwise sequentially Markovian caolescent (PSMC) analysis was used to reconstruct the demographic history of the NA populations here analyzed. Only data from highly covered regions were used. Reference population genomes were obtained from Complete Genomics (http://www.completegenomics.com/public-data/69-genomes/.) Bootstrapping analysis was performed to validate the inference as described by Li et al.[31] (Supplementary Fig. 9).

Secondly, MSMC models were used to analyze multiple genomes[33], grouped as follows: both individuals from each of the six NA groups of Mexico here analyzed; the four genomes from Northern NA populations of Mexico (Tepehuanos and Tarahumaras) and the four genomes from Southern Mexico (Zapotecas and Nahuas). All genomes were previously phased with SHAPEIT V2.12[65], using all individuals from 1KG project as reference panel. Demographic history of world populations was inferred using phased variants from the last release of the 1KG project.

**Potentially pathogenic alleles causing Mendelian disease.** The list of 56 genes recommended for reporting incidental findings by the ACMG was used as reference to identify potentially pathogenic variants[38]. Non-synonymous variants in the 12 NA genomes absent from the 1KG project (phase 3) were annotated using ClinVar database to identify potential pathogenicity.

**GWAS variants in NA individuals.** Based on the NHGRI GWAS Catalog containing all SNP-trait associations with a significance level of $10^{-5}$, we identified GWAS variants present in each genome[66]. Variants shared by the 12 NA individuals in heterozygous or homozygous form were grouped based on their EFO. The allele frequency of these variants was compared with that of other continental populations, particularly Europeans (CEU) and Asians (CHB and JPT) from the 1KG project (phase 3) (Supplementary Table 7). Thirteen variants were found in SNP 6.0 microarray and their allele frequencies were evaluated in 312 additional NA samples included in the present study (Supplementary Table 7).

**Pathway enrichment and gene ontology analysis**. Functional enrichment (Disease association and GO enrichment analysis) was performed using the WebGestalt online tool[67]. For disease association analysis, WebGestalt uses disease terms downloaded from PharmGKB and genes associated with individual disease terms were inferred using GLAD4U (Gene List Automatically Derived For You)[68]. Human genome data (hsapiens_genome) was used as reference set, the Benjamini-Hochberg multiple test was used to adjust for multiple testing[69]. Statistical significance was considered at $P < 0.01$. Three annotation matrix for enriched disease-associated genes were tested: (A) all genes with novel non-synonymous variants; (B) protein-coding genes with highest number of SNVs in the promoter region (above 95th percentile); and (C) genes from groups A and B combined. All analyses were performed seperately in each ethnic group.

**Data availability**. In consistency with the respective Institutional Review Board approval and individual informed consents, genome and exome variation files will be available at www.12g-data.inmegen.gob.mx, upon request to the corresponding authors.

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

## Acknowledgements

We thank the volunteers for their support for this research. We also thank Miguel Angel Contreras Siecks, Manuela Irina Ortega-Sánchez, Martha Lucía Granados-Riveros, and María Teresa Flores-Dorantes for assistance with sample processing and data analyses; Stephan Schiffels and Rolando González-José for comments on the manuscript; and Alejandro Rodríguez-Torres for his help in Fig. 1. This work was supported by funds from the Federal Government of Mexico to the Instituto Nacional de Medicina Genómica.

## Author contributions

Conceived and designed study: S.R.-H., A.O.-L., A.G., V.A.-A., S.C.-Q., X.S. Contributed reagents/materials: V.A.-A., E.A.-A., M.B.-Q., R.B.-L., H.G.-O., A.G.-S., J.G., F.G.-R., P.L.-M., G.M.-P., A.M.-H., M.M., L.O., G.O.-L., J.R.-M., H.V.-R., T.V.-M. Performed the experiments: S.R.-H., A.O.-L., A.G. Analyzed data: S.R.-H., A.O.-L., A.G., A.C., F.C.-G., J.C.F.-L., H.G.-O., E.H.-L., E.M., F.P.-V., F.R.-M, M.V.-C. Wrote the paper, incorporating input from all authors: S.R.-H., A.O.-L., A.G., T.V.-M., S.C.-Q., X.S.

## Additional information

**Competing interests:** The authors declare no competing financial interests.

