## [Peer Review File · Nature Communications]

Reviewers' comments:

Reviewer #1 (Remarks to the Author):

Soberon et al. analyze the high coverage WGS of 12 Native Mexican and array-genotyping data of 320 individuals, all these Native Americans, with different ethnic groups. The authors explore the genetic structure of Mexican Natives. It also analyze the pattern of rare variants identified by WGS. The manuscript is well organized and presented and I recommend to publish it after revision.

My major concern is that array data from the 320 individuals should be better presented. The manuscript gains in robustness if present both type of data and taking advantage of the characteristics of each dataset. For example, Admixture analyses should be based more in array-data, while GWAS-hits allele frequency comparison would be performed using the array data.

Additional points are presented as comments in the attached word file.

Reviewer: Eduardo Tarazona-Santos

Reviewer #2 (Remarks to the Author):

Romero-Hidalgo et al., present American Indian genomes with some biologically relevant findings.

This is one of the very first locally submitted population genome analyses papers. They cover the common genome aspects of the indigenous populations.

To this reviewer, the most interesting biological aspect is the novel misense and promoter region variants as some Amerindian populations show an extreme level of endurance.

The authors performed a proper and thorough genomes analyses with high enough sequencing depth data.

This reviewer finds that all the analyses are of high quality with good analysis plans.

I think it deserves to be published promptly.

The authors included Anzick-1 genome which is quite useful for this analysis.

Reviewer #3 (Remarks to the Author):

This paper reports analyses based on 15 Mexican individuals whole-genome sequenced to high coverage. Using these data the authors performed a range of bioinformatic analyses in order to explore population genetic and disease annotation questions. The data appear of good quality and the analyses seem to have been performed adequately. My main overall concern with the paper is the lack of novelty with regards to most of the results. This relates particularly to the population genetic analyses (e.g. the patterns of sequence diversity observed are similar to those seen in other Native American sequences and the varying diversity of the Mexican populations examined has been mostly reported in other genetic surveys). With regards to the gene annotation analyses, these lead to what

can only be considered speculations regarding the possible biological significance of the sequence variation detected. Thus, the paper oscillates between the description of rather unsurprising results and speculative statements about the variant annotation analyses. For instance, it is stated that the apparent enrichment, in the Tarahumaras, of new non-synonymous/promoter variants in gene pathways involved in musculo-skeletal diseases “could be related to the well-known high physical resistance of this Native Mexican group” (bottom of page 12). Assuming that this enrichment is real (ignoring the small number of Tarahumara examined, and the fact that in Table 4 many other populations also show significant values) and also assuming that the Tarahumara do have this distinctive physiological feature, there is a huge leap between such genome annotations and physiology. To start with, the genome annotations examined relate to disease associations and it is unclear how these diseases might relate to physiology. In that respect, a more closely related data would be, for instance, if the Tarahumara had a high prevalence for the musculoskeletal diseases included in the annotations. And even then, direct testing of such an association would be required and ultimately a biological explanation. Broadly, this sort of speculation should therefore be majorly toned down throughout the paper.

I include a number of specific comments in the manuscript pdf. A prominent presentation issue is that the figures are not well integrated with the text. Some of these (2,4) seem rather unnecessary. The legends to the figures are also disjointed with respect to the main text, as if they had been prepared quite independently (e.g. the title of Fig 5 includes “PharmGKB enrichment analysis” but PharmGKB is not mentioned at all in the main text).

Response to Reviewers' comments

Whole genome sequencing of Native Americans of Mexico provides new insights on demographic history and biologically relevant genetic variation

Reviewer #1

Thank for reviewing our manuscript and for your comments.

1) *“My major concern is that array data from the 320 individuals should be better presented. The manuscript gains in robustness if present both type of data and taking advantage of the characteristics of each dataset. For example, Admixture analyses should be based more in array-data, while GWAS-hits allele frequency comparison would be performed using the array data.”*

Response: According to your suggestion, we re-analyzed the micro-array data for the admixture analysis in Native Americans, including the data published by Moreno-Estrada et al. 2014 and the Native Americans included in this study. This allowed us to increase SNP density for the admixture analysis. Results were very similar to those described in the first version of our manuscript, and are now included in the revised version.

In addition, as you suggested we analyzed the allele frequencies of GWAS-hits observed from the analysis of the 12 genomes in the additional 312 Native Americans presented in this study. Of the 44 variants with the highest allele frequency differences between Europeans, Asians and the 12G, only 13 were found in the microarray genotypes of the 312 NA samples. Allele frequencies of these 13 SNVs were compared between the 312 NAT samples and the 12G, and were consistently more frequent in Native Americans. These data are now included in Supplementary Table 7.

Reviewer #2

Thank you for your kind comments.

Reviewer #3

Thank for reviewing our manuscript and for your comments.

1. *“... it is stated that the apparent enrichment, in the Tarahumaras, of new non-synonymous/promoter variants in gene pathways involved in musculo-skeletal*

diseases “could be related to the well-known high physical resistance of this Native Mexican group” (bottom of page 12). Assuming that this enrichment is real (ignoring the small number of Tarahumara examined, and the fact that in Table 4 many other populations also show significant values) and also assuming that the Tarahumara do have this distinctive physiological feature, there is a huge leap between such genome annotations and physiology. To start with, the genome annotations examined relate to disease associations and it is unclear how these diseases might relate to physiology. In that respect, a more closely related data would be, for instance, if the Tarahumara had a high prevalence for the musculoskeletal diseases included in the annotations. And even then, direct testing of such an association would be required and ultimately a biological explanation. Broadly, this sort of speculation should therefore be majorly toned down throughout the paper. “

Response: We agree that gene enrichment in pathways involved in musculoskeletal diseases is not conclusive evidence of the physical resistance of this group, and that there is a huge leap between such genome annotations and physiology. Therefore, the statements regarding this issue were toned down throughout the article and the statement referring to this issue in the abstract was removed.

2. Regarding the statement “*assuming that the Tarahumara do have this distinctive physiological feature*”,

Response: There is in fact documented evidence of the remarkable physical endurance of Tarahumaras (Balke and Snow, 1965; Groom, 1971; Irigoyen, 1985)

Regarding the statement “*a more closely related data would be, for instance, if the Tarahumara had a high prevalence for the musculoskeletal diseases included in the annotations*”,

Response: To our knowledge there is no evidence of a high frequency of musculoskeletal disease in Tarahumaras. However, the enrichment of gene variants in genes associated with musculoskeletal and bone diseases/abnormalities does not mean that the variants found in Tarahumaras will be necessarily detrimental and associated with disease. For example, COL5A1 polymorphisms have been associated with running performance in ultra-marathons (Brown JC et al., 2011; Abrahams S, 2014), while in other studies COL5A1 polymorphisms have been associated with risk of diseases such as Achilles tendon pathology (Brown KL et al., 2017) and carpal tunnel syndrome (Dada S et al., 2016).

Regarding the statement “*And even then, direct testing of such an association would be required and ultimately a biological explanation.*”

Response: We agree that this analysis in no way confirms that this variation is involved in physiology related to high physical resistance, and thus the speculation was toned down throughout the manuscript. Unfortunately, testing for associations of variants with high physical resistance in this population is currently unfeasible. However, in order to test whether this enrichment pattern persisted in other Tarahumaras, we sequenced the exome of 3 more Tarahumaras who had previously participated in high resistance physical activity events. We found

enrichment patterns consistent with those of the two initially sequenced Tarahumaras. These results have been included in the revised version of the manuscript.

3. A prominent presentation issue is that the figures are not well integrated with the text. Some of these (2,4) seem rather unnecessary. The legends to the figures are also disjointed with respect to the main text, as if they had been prepared quite independently (e.g. the title of Fig 5 includes “PharmGKB enrichment analysis” but PharmGKB is not mentioned at all in the main text).

Response: We agree that the figures were not well integrated in the text. We made several changes to address this issue. In addition, according to your suggestion, figures 2 and 4 were removed from the main manuscript and included as supplementary figures (Supplementary Fig 2 and 3). We also reviewed figure legends, removed “PharmGKB enrichment analysis” from the title of figure 4, and clarified issues concerning the enrichment analysis in the methods section.

Specific comments throughout the manuscript pdf:

- a) The introduction section was modified according to your suggestions.
- b) In the results and discussion section, we made several modifications according to your suggestions. We included a map describing the location of each population. We also described heterozygosity in all ethnic groups and relatedness estimates (IBD) between both individuals of each group. We fully agree that the reconstruction of the demographic history was unclear, and have made modifications hoping it is now clear.
- c) Figure 2 (Supplementary Fig 2 in the revised manuscript): The purpose of the Treemix analysis is now described in the text. The position of the Karitiana (a Native American from Brazil) is as expected according to other publications (Rasmussen et al. 2014; Raghavan et al. 2015). The arrows (yellow and red) indicate gene flow between lineages; this is described in the figure legend.
- d) Figure 3: In the text we now refer to an extended bottleneck which is consistent with estimates of the time that the Native American ancestors crossed the Bering Strait and moved into America.

REVIEWERS' COMMENTS:

Reviewer #1 (Remarks to the Author):

The authors have addressed most of the recommendations formulated by the reviewers in the previous version, but in my opinion, few important issues persist that should be fixed.

Specifically:

1) For ADMIXTURE analyses, please estimate and include the cross-validation error graph and show that $K=10$ correspond to the lowest (or is associated to a low) CV error. Also, in the result section it would be better to change "structure analysis" by "ADMIXTURE analysis", because "structure analysis" may give the impression that analyses were performed with the Structure software (or its new faster version).

2) specify which relatedness measurement are you using. I suggest the classical kinship coefficient.

3) also, correct the term heterozygosity

All these comments are inserted in the pdf version as comments.

Eduardo Tarazona Santos

Reviewer #3 (Remarks to the Author):

The authors have incorporated most of my suggested comments. I find that the manuscript has improved substantially and is ready for publication. I have no further comments.

In response to the issues raised by reviewer 1:

1. *“For ADMIXTURE analyses, please estimate and include the cross-validation error graph and show that K=10 correspond to the lowest (or is associated to a low) CV error. Also, in the result section it would be better to change “structure analysis” by “ADMIXTURE analysis”, because “structure analysis” may give the impression that analyses were performed with the Structure software (or its new faster version).”*

Response: We have included the cross-validation error graphs, and made the appropriate changes in the manuscript. We changed “structure analysis” for “admixture analysis” throughout the text.

2. *“Specify which relatedness measurement are you using. I suggest the classical kinship coefficient”.*

Response: According your suggestion, we used kinship coefficient to estimate relatedness using the KING v2.0 program. The appropriate changes have been made in text and methods section. The Totonacos were the only group showing relatedness (third degree).

3. *“correct the term heterozygosity”.*

Response: The term has been corrected in the manuscript.

Regarding the issues raised on the pdf document, we have reduced the title and abstract, and considered all corrections. Only a few issues deserve clarification:

- a. Nahua is a Native American population, Nahuatl is a language. It is best to leave the phrase as it is.
- b. The treemix analysis included data from the 12 NA genomes; this mistake has been corrected in the results and discussion sections.
- c. In response to your request, we entitled the section “Results” and included a “Discussion” section with conclusions, similar to other articles published in Nature Communications (e.g. Adhikari, K. *et al.* A genome-wide association study identifies multiple loci for variation in human ear morphology. *Nat Commun.* **6**, 7500; 2015).
- d. The map was designed at INMEGEN, it was not designed from a stock or 3rd party image, no copyright permission is required.
- e. In consistency with the respective Institutional Review Board approval and individual informed consents, genome and exome variation files will be

available at www.12g-data.inmegen.gob.mx, upon request to the corresponding authors. This statement has been included in the manuscript.